# Climate-dependent propagation of precipitation uncertainty into the water cycle

Ali Fallah[1,2], Sungmin O[2], and Rene Orth[2]

[1]Department of Civil and Environmental Engineering, Shiraz University, Shiraz, Iran

[2]Department of Biogeochemical Integration, Max Planck Institute for Biogeochemistry, D-07745, Jena, Germany

*Correspondence to*: Ali Fallah (alifallah@shirazu.ac.ir; afallah@bgc-jena.mpg.de)

**Abstract.** Precipitation is a crucial variable for hydro-meteorological applications. Unfortunately, rain gauge measurements are sparse and unevenly distributed, which substantially hampers the use of in-situ precipitation data in many regions of the world. The increasing availability of high-resolution gridded precipitation products presents a valuable alternative, especially over poorly gauged regions. This study examines the usefulness of current state-of-the-art precipitation datasets in hydrological modelling. For this purpose, we force a conceptual hydrological model with multiple precipitation datasets in >200 European catchments to obtain runoff and evapotranspiration. We consider a wide range of precipitation products, which are generated via (1) interpolation of gauge measurements (E-OBS and GPCC V.2018), (2) data assimilation into reanalysis models (ERA-Interim, ERA5, and CFSR), and (3) combination of multiple sources (MSWEP V2). Evaluation is done at the daily and monthly time scales during the period of 1984-2007. We find that simulated runoff values are highly dependent on the accuracy of precipitation inputs, by contrast, simulated evapotranspiration is generally much less influenced in our comparatively wet study region. We also find that the impact of precipitation uncertainty on simulated runoff increases towards wetter regions, while the opposite is observed in the case of evapotranspiration. Finally, we perform an indirect performance evaluation of the precipitation datasets by comparing the runoff simulations with streamflow observations. Thereby, E-OBS yields the particularly strong agreement, while furthermore ERA5, GPCC V.2018 and MSWEP V2 show good performance. We further reveal climate-dependent performance variations of the considered datasets, which can be used to guide their future development. The overall best agreement is achieved when using an ensemble mean generated from all the individual products. In summary, our findings highlight a climate-dependent propagation of precipitation uncertainty through the water cycle; while runoff is strongly impacted in comparatively wet regions such as Central Europe, there are increasing implications on evapotranspiration towards drier regions.

## 1. Introduction

Precipitation is a key quantity in the water cycle since it controls water availability including both blue and green water resources (Falkenmark and Rockström, 2006; Orth and Destouni, 2018). This way, changes in precipitation translate into changes in water resources which could have severe impacts on ecosystems, and consequently economy and society (Oki and Kanae, 2006; Kirtman et al., 2013; Abbott et al., 2019). Changes in precipitation can be induced or intensified by climate change and consequently lead to amplified impacts (Blöschl et al., 2017; Blöschl et al., 2019b). Thus, accurate precipitation information is essential for monitoring water resources and managing related impacts.

Despite the necessity of accurate precipitation datasets, in most regions, reliable gauge measurements are not widely available. Further, these measurements need to be corrected for potential errors such as wind-induced inaccuracies or precipitation undercatch, especially in higher altitudes (Sevruk et al., 2009; Mekonnen et al., 2015; Zandler et al., 2019; Duethmann et al., 2020). Next to gauge measurements, precipitation information can be inferred from satellite observations and/or model simulations. Based on these sources, a variety of global gridded precipitation datasets have emerged. While some of these datasets make direct use of gauge measurements to interpolate them in time and space, others make indirect use of the gauge information to calibrate satellite retrieval algorithms or models, enabling them to estimate gridded large-scale precipitation.

Across these datasets, there are ample discrepancies in space and time, highlighting the need for comparative assessments (e.g. Koutsouris et al., 2016; Alijanian et al., 2017, 2019; Balsamo et al., 2018; Sun et al., 2018; Massari et al., 2019; Brocca et al., 2019; Sharifi et al., 2019; Caroletti et al., 2019; Levizzani and Cattani, 2019; Roca et al., 2019; Fallah et al., 2020; Satgé et al., 2020; Contractor et al., 2020; Xu et al., 2020; Zhou et al., 2020). In particular, indirect evaluation of the datasets through application in hydrological modelling is a valuable alternative in this context as precipitation is translated into variables with more reliable observations such as runoff, as long as this is measured in catchments with near-natural dynamics (Thiemig et al., 2013; Nerini et al., 2015; Beck et al., 2017a,b,2019a,b; Fereidoon et al., 2019; Bhuiyan et al., 2019; Mazzoleni et al., 2019; Arheimer et al., 2020; Dembélé et al., 2020). However, while this approach relies on the propagation of precipitation uncertainty into runoff it is largely underexplored when and where this propagation pathway is active. Vice versa, it is unclear in which regions or conditions, gridded datasets of runoff (Gudmundsson and Seneviratne, 2016) or evapotranspiration (e.g. Martens et al., 2017; Jung et al., 2019) are impacted by the existing precipitation uncertainties.

In this study, we investigate the uncertainty across six widely used gridded precipitation datasets, including its propagation into the hydrological cycle, i.e. runoff and evapotranspiration (ET). Thereby, we consider gauge-interpolated (E-OBS v17.0, GPCC V.2018), reanalyses (ERA-Interim, ERA5, CFSR), multi-source (MSWEP V2) datasets. With each of them, and with an ensemble mean computed from all of them, we force a conceptual land surface model and compare the respectively simulated runoff and ET. This is done separately for different hydro-climatological regimes. In addition, validating the runoff simulations against respective observations we can indirectly infer the performance of the precipitation datasets. This further allows us to obtain guidelines with respect to the usefulness of the different types of precipitation products in the considered regimes.

Section 2 introduces the reference, forcing datasets and model calibration used in the study, and Section 3 illustrates results and discussion. Finally, in Section 4 the conclusions of this study are presented.

## 2. Data and methodology

### 2.1. Forcing data

Runoff and ET are modelled with a conceptual hydrological model, the Simple Water Balance Model (SWBM). The underlying framework was initially presented by Koster and Mahanama (2012) where runoff (normalised by precipitation) and ET (normalised by net radiation) are assumed to be polynomial functions of soil moisture (Whan et al., 2015). We use here the model version introduced by Orth and Seneviratne (2015) in which the original model is adapted to the daily time scale by addition of an implicit form of the water balance equation and a streamflow recession parameter which enables streamflow that is delayed with respect to the respective precipitation event. Please refer to Orth and Seneviratne (2015) for the relevant model equations and validation results. Note that the basic concept and the governing equations of runoff and ET formation used here are well established and

employed in many similar conceptual models, such as HBV (Bergström 1995; Orth and Seneviratne, 2015). As inputs, the model uses temperature, net radiation, and precipitation. For each catchment, temperature and net radiation are used from the respective grid cells from the E-OBS (Cornes et al., 2018) and ERA-Interim (Dee et al., 2011) datasets, respectively. Corresponding grid cell-based precipitation data is used from various datasets derived from different sources: gauge-based (E-OBS, GPCC V.2018), reanalysis (ERA-Interim, ERA5, CFSR), and multi-source datasets (MSWEP V2). A summary of all precipitation datasets and their respective characteristics is shown in Table 1.

Before using the precipitation datasets to force the SWBM, they are re-gridded to a common 0.5° spatial resolution, if necessary. This was done through conservative remapping which preserves the water mass (Jones, 1999) using climate data operators (Schulzweida, 2019). The SWBM simulations are performed with a daily time step, and the analysis thereof is done at daily and monthly time scales.

## 2.2. Reference data

Modelled runoff is evaluated against streamflow observations obtained from 416 catchments distributed across Europe (Stahl et al., 2010). The streamflow data were collected from the European water archive, national ministries and meteorological agencies and from the WATCH project. These daily data are available for the period 1984-2007. There is no or little human influence on the streamflow in these catchments, which are mostly between 10-100 km$^2$ in size. More details on the data and catchments can be found from Stahl et al., 2010.

## 2.3. Model calibration

The simple water balance model employed in this study includes six adjustable parameters: water-holding capacity, inverse streamflow recession time scale, runoff ratio exponent, ET ratio exponent, maximum evaporative fraction, and a snow melting parameter (as in Orth and Seneviratne 2015, see also Table S1). For model calibration, 500 parameter sets are tested which are randomly sampled from the entire parameter space using Latin Hypercube Sampling (LHS; McKay et al., 1979). The ranges for each parameter within this parameter space are obtained from O et al., 2020 (see also Table S1). This way, we performed 500 corresponding simulations for each catchment over the entire considered time period 1984-2007. For each simulation, we computed the resulting Nash-Sutcliffe efficiency (NSE, Nash and Sutcliffe, 1970) between observed and simulated runoff to determine the best-performing parameter set. The results are shown in Figure S1. In addition, any catchments with NSE<0.36 in the case of the best parameter set were disregarded from the further analyses. This NSE threshold for the catchment selection is adopted from Motovilov et al. (1999) and Moriasi (2007). The model was deemed not applicable there due to e.g. human influence on the local runoff dynamics, or model shortcomings. This way, out of the original >400 catchments, 264 are retained for the actual analyses, which are well distributed across the European continent and its climate regimes.

Note that we perform only calibration of the model, and no validation. This is because we focus on the influence of the precipitation forcing on the modelled runoff performance, and not on the simulation capacity of the model outside training conditions. Satisfactory predictive performance of the model has been shown in previous studies (Orth et al., 2015; Orth and Seneviratne, 2014, 2015; Schellekens et al., 2017; O et al., 2020).

As shown in Fig. 1, the hydro-climatological regime is characterised through long-term average temperature and aridity (Budyko, 1974). Thereby, for each catchment, the temperature is taken from the E-OBS dataset, and aridity is computed as the ratio of mean annual net radiation to mean annual precipitation calculated from ERA-Interim and E-OBS, respectively.

In each of the 264 catchments, the SWBM is forced with temperature, net radiation and the different precipitation datasets, respectively, as illustrated in Figure 2. This way, six simulations with the six different precipitation datasets are performed for each catchment, leaving the temperature and net radiation data unchanged. The model parameters are thereby obtained from the above-mentioned calibration using E-OBS precipitation. As this can potentially introduce biases into our results, we additionally calibrated the model using GPCC V.2018 precipitation data to derive alternative parameters with which we re-computed the main analyses.

All analyses are performed during the warm season (May-September) to minimise the impact of snow and ice even though snow melting can locally affect streamflow even in the warm season (Jenicek et al., 2016).

## 3. Results and discussion

### 3.1. Impact of precipitation uncertainty on runoff and ET

Figure 3 illustrates the propagation of precipitation uncertainty into simulated runoff and ET at the monthly scale. Each point denotes the standard deviation across the six simulations obtained with the different precipitation datasets and represents a particular month (May-Sep) in a specific catchment. Runoff simulations are impacted by precipitation uncertainty while the ET simulations are much less influenced by precipitation uncertainty, as indicated by the regression slope. The clear relationship between runoff and precipitation is in line with previous studies (e.g. Beck et al., 2017a,b; Sun et al., 2018, Blöschl et al., 2019b). It is related to the fact that most of the considered catchments are located in relatively wet climate (aridity<1) such that soils are often saturated, triggering a rather direct runoff response to precipitation (Ghajarnia et al., 2020). Also, in these climate regimes ET is typically energy-controlled rather than water-controlled (Koster et al., 2004; Zheng et al., 2019; Pan et al., 2020; Denissen et al., 2020), leading to the observed low sensitivity of ET to precipitation (uncertainty).

### 3.2. Climate-dependent propagation of precipitation uncertainty

In addition to examining the role of precipitation uncertainty for runoff and ET across all considered catchments, we analyse this uncertainty propagation within individual hydro-climatological regimes (Fig. 4). For this purpose, we compute the median of the standard deviations from catchments within each regime, considering all respective warm season months. Figure S2 shows the number of catchments located within each regime. Only regimes with >5 catchments are considered in the analysis. The uneven distribution of catchments across the regimes induces higher uncertainties in the results obtained for the wettest and driest regimes. As shown in Fig. 4a, the precipitation variability across the considered products is higher in comparatively cold and wet regions. This could be related to especially sparse gauge networks and more intense rainfall in these regions which are known to increase precipitation uncertainty (Dinku et al., 2008; Hu et al., 2016; Beck et al., 2017b; O and Kristetter, 2018).

Similarly, Figs. 4b and 4c illustrate the fraction of precipitation uncertainty propagating into runoff and ET, respectively. Interestingly, we find systematic variations in this uncertainty propagation with respect to climate. In wet and cold regions, precipitation uncertainty almost exclusively affects runoff whereas ET remains unchanged. Towards drier and warmer climate the uncertainty propagation shifts, affecting runoff less and increasingly influencing ET.

In addition to the previous analyses using monthly averaged data, we re-compute Figure 4 using daily data. The results are shown in Figure S3. The similarity between Figures 4 and S3 suggests that our findings on the climate-dependent propagation of precipitation uncertainty are valid across daily and monthly time scales. Further, we repeat the uncertainty propagation analysis

using (i) model parameters obtained from calibration with GPCC V.2018 precipitation forcing instead of E-OBS precipitation (Figure S4), (ii) using all months instead of focusing on the warm season (Figure S5), and (iii) using an NSE limit of 0.5 instead of 0.36 to select catchments where the SWBM is applicable (Figure S6). We find that Figures S4-S6 show similar patterns as in Figure 4, which confirms that our findings are robust with respect to the methodological approach, particularly in terms of the precipitation dataset employed for model calibration, the considered season, and the applied NSE threshold to determine the applicability of the model (see also Section 2.3).

### 3.3. Indirect validation of precipitation dataset qualities

Given the preferential propagation of precipitation uncertainty to runoff in the considered European catchments, we focus in the following on runoff only. In this context, we use streamflow measurements from the catchments to validate the modelled runoff, which allows us to draw conclusions also on the usefulness of the employed precipitation forcing datasets, and of a mean ensemble thereof. This is, however, not possible in the case of ET due to the lacking relationship between ET and the precipitation forcing in our study region (Figure 3). For the runoff validation, we consider the correlation of monthly anomalies in each catchment and the absolute bias. To obtain anomalies, we remove the mean seasonal cycle from the observed and modelled runoff time series of each catchment. The six runoff simulations derived with the individual precipitation products alongside the runoff simulation obtained with the mean ensemble are then ranked in each catchment with respect to (i) correlation and (ii) bias. The sum of these 2 ranks is used to obtain an overall ranking of runoff simulations and corresponding precipitation forcing datasets in each catchment.

Figure 5 shows the number of catchments in which each precipitation product yields the best-ranked runoff simulation. Our findings show that overall the performance of modelled runoff is clearly dependent on the employed precipitation product. This is expected given the considerable disagreement between precipitation products, and the preferential propagation of this uncertainty to runoff (Fig. 4). Generally, among the individual products, the runoff computed with E-OBS precipitation agrees best with observations. Also, ERA5, MSWEP V2, and GPCC V.2018 yield comparatively good runoff estimates. In contrast, runoff simulations obtained with ERA-Interim and CFSR agree less well with observations. The precipitation ensemble outperforms all individual products, highlighting the usefulness of multi-source and multi-product approaches in the derivation of suitable precipitation datasets for hydrological modelling. Furthermore, we compute runoff performance assessments separately for anomaly correlation and absolute bias (Fig. S10). This reveals that the performance of the precipitation datasets is rather similar in terms of resulting runoff biases. Only ERA5 seems to lead to reduced biases compared with the other products, probably as it does not suffer from a gauge-based precipitation undercatch. In contrast, there are considerable differences in terms of the runoff anomaly correlation performance across the products. This indicates that the differences across products shown in Fig. 5 are mostly resulting from contrasting performance with respect to runoff anomaly correlation.

Repeating the evaluation from Figure 5 with daily data (Figure S7) we find similar results. This suggests that the relative quality of the considered precipitation is comparable across daily and monthly time scales. In addition, we re-compute Figure 5 using all months of the year (Fig. S8), and GPCC-derived SWBM parameters (Fig. S9), which both largely confirm the described results. Note that, not surprisingly, model calibration with a particular precipitation product, e.g. E-OBS or GPCCV.2018, leads to the better performance of that respective product.

Figure 6 shows the runoff performance resulting from the various precipitation products for the previously considered hydro-climatological regimes. Interestingly, we find remarkable performance differences across the regimes, suggesting differential usefulness of precipitation products for hydrological modelling across different climate zones. Also, we can identify regimes where the precipitation products perform particularly well or not. For example, MSWEP V2 leads to strong agreement between modelled

and observed runoff mostly in comparatively cold and wet climate and less so in warmer and drier regimes. This might be related to problems of the products incorporated in MSWEP in capturing convective rainfall in warm and dry regions while this is less problematic in colder regions (Ebert et al., 2007; Beck et al., 2017a,b; Massari et al., 2017; Fallah et al., 2020). The opposite performance pattern is observed for GPCC V.2018. The weaker performance in cold climate, which is also present in the case of E-OBS, might be related to smaller gauge network density, and more complex topography in colder areas (Beck et al., 2017b; Ziese et al., 2018). For the other products such as CFSR and ERA-Interim, the performance is less dependent on the hydro-climatological regime.

## 4. Conclusions

In this study, we investigate how the remarkable discrepancy across state-of-the-art gridded precipitation datasets propagates through the water cycle. This is essential for hydrological modelling and the applicability of resulting simulations of water balance components such as runoff or ET. Our findings reveal that the uncertainty across precipitation datasets propagates mainly into runoff rather than ET simulations in Europe. In addition, the partitioning of precipitation uncertainty between runoff and ET is climate-dependent. In comparatively cold and wet regions such as Europe runoff is more impacted, whereas in drier and warmer regions the uncertainty partitioning shifts towards ET. This applies across daily and monthly time scales.

The results in this study are obtained with a single model and are potentially dependent on the choice of that model. Even though this model has been validated thoroughly and applied in previous studies (Orth et al., 2015; Orth and Seneviratne, 2014, 2015; Schellekens et al., 2017; O et al., 2020), future research needs to explore precipitation error propagation with other models (as in Bhuiyan et al., 2019). This should particularly include distributed models adding to our use of a lumped scheme. However, we do obtain similar results with different calibrations of this model, while previous research indicated that differences across model calibrations can be similar to that across models (Tebaldi and Knutti, 2007).

The strong link between precipitation and runoff in Europe allowed us to perform an indirect validation of precipitation products through the performance of the respectively modelled runoff. Overall, the E-OBS precipitation dataset yields the most reliable streamflow simulations in Europe across the considered precipitation products. Weaker but still comparatively good agreement between modelled and observed streamflow is obtained with ERA5, GPCC V.2018 and MSWEP V2. Thereby the products differ mostly with respect to the temporal dynamics rather than the overall amount of precipitation (Sun et al., 2018; Fallah et al., 2020). The interpolated products overall outperform the satellite-derived products in Europe. This is probably due to the high density of gauge observations, as previous research found contrasting conclusions in regions with low gauge density (e.g. Thiemig et al., 2013 for Africa). We further find that the ensemble mean of the considered precipitation datasets outperforms the individual datasets, suggesting that such approaches are promising to obtain more reliable precipitation forcing for hydrological modelling as shortcomings in individual datasets seem to cancel out to some extent when used within an ensemble. Further, we study the performance of the considered precipitation products with respect to climate. We find systematic variations for datasets like MSWEP V2 and GPCC V.2018 whereas ERA5, ERA-Interim, and CFSR perform more similarly across climate regimes. Revealing climate-dependent accuracies in some precipitation datasets supports focused development of these products. This way, innovative hydrological validation of precipitation data, in addition to direct validation against ground truth, can contribute to address the still considerable uncertainty across state-of-the-art gridded products in future efforts.

Further, these findings allow a more targeted combination of products to compensate for individual weaknesses and preserve respective strengths. The climate-dependent (propagation of) precipitation uncertainties illustrates that there is no best overall product but instead a careful regional, climate-based selection can support hydrological applications. Overall, these findings highlight the usefulness of streamflow measurements capturing truly large-scale hydrological dynamics which can then be used to make inference on the accuracy of precipitation datasets (Behrangi et al., 2011; Thiemig et al., 2013; Beck et al., 2017a, 2019a; Bhuiyan et al., 2019; Mazzoleni et al., 2019; Alnahit et al., 2020; Arheimer et al., 2020).

Another important outcome of our analyses is that ET simulations are mostly insensitive to precipitation uncertainty in European climate, confirming previous studies (Bhuiyan et al., 2019; Zheng et al., 2019). However, in warmer and drier regions such as the Middle East, Central North America or Australia, the link between ET and precipitation should be stronger. Wherever available in these regions, ET measurements can and should be used for indirect evaluation of large-scale precipitation products to complement the results in this study where we focused more on comparatively wet regions.

Moreover, our findings suggest that, across Europe and regions with similar climate, gridded runoff datasets (e.g. Gudmundsson and Seneviratne, 2016) inevitably suffer from the existing uncertainty in state-of-the-art precipitation datasets, although this depends on the extent to which they rely on precipitation data. In contrast, gridded ET products (e.g. Martens et al., 2017, Jung et al., 2019) are not impacted by precipitation uncertainty in these regions. In warmer and drier regions, however, the gridded ET products are more challenged than the runoff products.

Overall, our findings highlight the important role of precipitation accuracy and the understanding of the propagation of existing inaccuracies through the water cycle. Revealing the climate-dependency of this propagation, this study contributes to improved modelling and monitoring of water resources which is of particular relevance in the case of extreme events such as floods and droughts (e.g. Golian et al., 2019; Alexander et al., 2020), which might increase in a changing climate.

*Competing interests.* The authors declare no conflicts of interest.

*Acknowledgment*. The authors thank the anonymous reviewers for their valuable comments. Further, we appreciate the assistance of Ulrich Weber for preparing the precipitation datasets. Ali Fallah acknowledges financial support from Ministry of Science, Research and technology of the I.R. of Iran, and also the support in the form of hosting and supervision provided by the Max Planck Institute for Biogeochemistry in Jena, Germany. Rene Orth and Sungmin O acknowledge funding support by the German Research Foundation (Emmy Noether grant number 391059971). We acknowledge the E-OBS dataset from the EU-FP6 project UERRA (http://www.uerra.eu) and the Copernicus Climate Change Service, and the data providers in the ECA&D project (https://www.ecad.eu). Also, we acknowledge GPCC V.2018 [https://opendata.dwd.de], MSWEP V2 [http://www.gloh2o.org], ERA-Interim [https://www.ecmwf.int/en/forecasts/datasets/reanalysis-datasets/era-interim], ERA5[https://climate.copernicus.eu/climate-reanalysis] and CFSR [https://cfs.ncep.noaa.gov]). Further, we are thankful for streamflow data from a dataset compiled by Stahl et al., 2010, who collected data from the European water archive ([http://www.bafg.de/GRDC/], accessed 9 December 2019), from national ministries and meteorological agencies, as well as from the WATCH project ([http://www.eu-watch.org], accessed 9 December 2019).

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

**Table 1: Summary of the precipitation datasets evaluated in this study**

| Group | Dataset | Temporal coverage | Spatial coverage | Spatial resolution | Data sources | Reference |
|---|---|---|---|---|---|---|
| **Interpolated** | E-OBS | 1950-2019 | Europe | 0.25° | Gauge | Cornes et al., 2018 |
| | GPCC V.2018 | 1901-2016 | Global | 1° | Gauge | Ziese et al., 2018 |
| **Multi-source** | MSWEP V2 | 1979-2017 | Global | 0.1° | Satellite + Gauge + Reanalysis | Beck et al., 2019 |
| | ERA-Interim | 1979-2019 | Global | 0.5° | Reanalysis | Dee et al., 2011 |
| **Modelled** | ERA5 | 1950-current | Global | ~0.28° | Reanalysis | Copernicus Climate change Service, 2017 |
| | CFSR | 1979-current | Global | 0.5° | Reanalysis | Saha et al., 2010, 2012 |

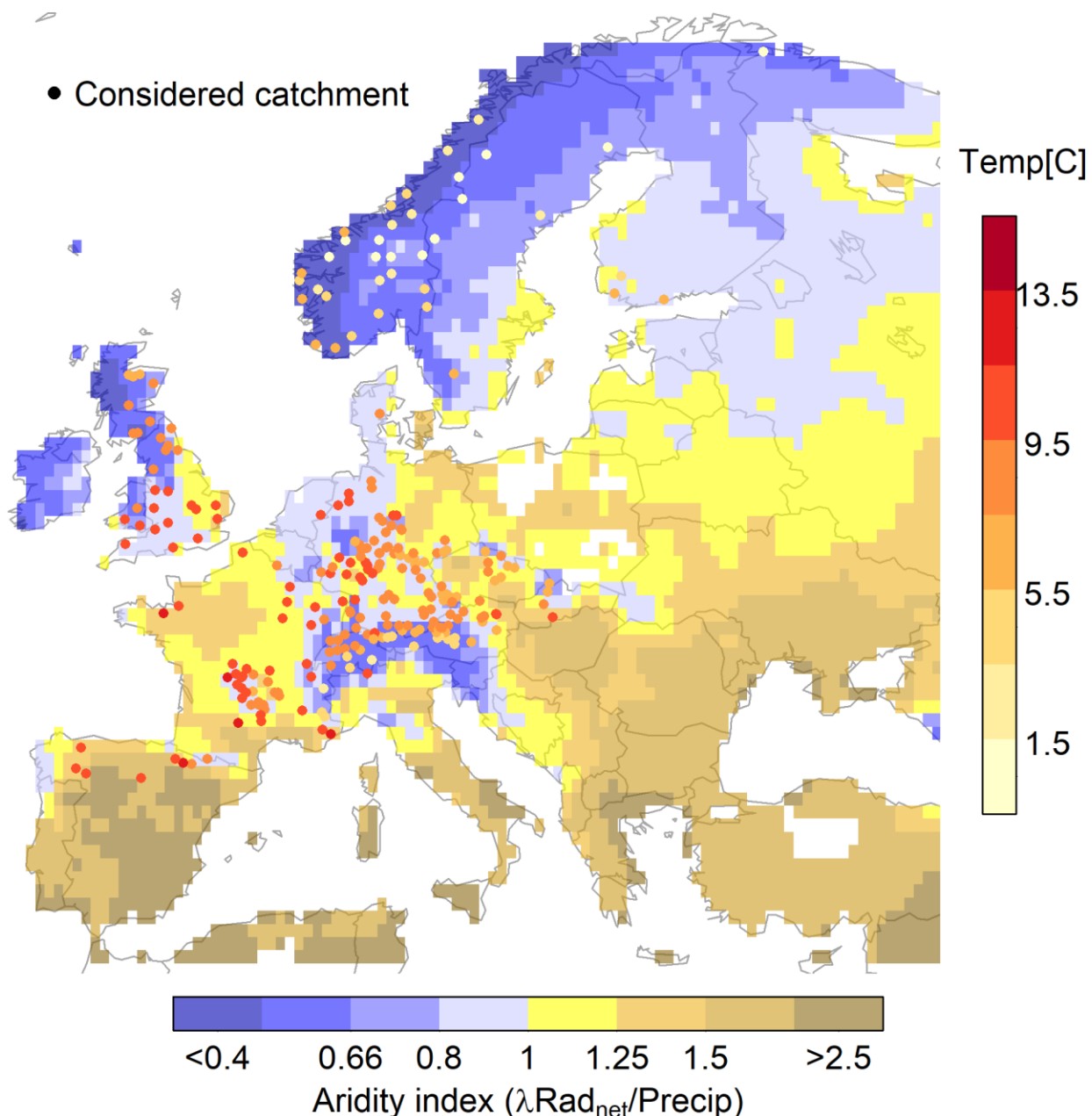

**Figure 1: Map of the study area. Signs mark the position of the 264 study catchments, with color indicating their annual average temperature. Map colors show the aridity index of regions as determined by a ratio of long-term average net radiation and precipitation (1984-2007).**

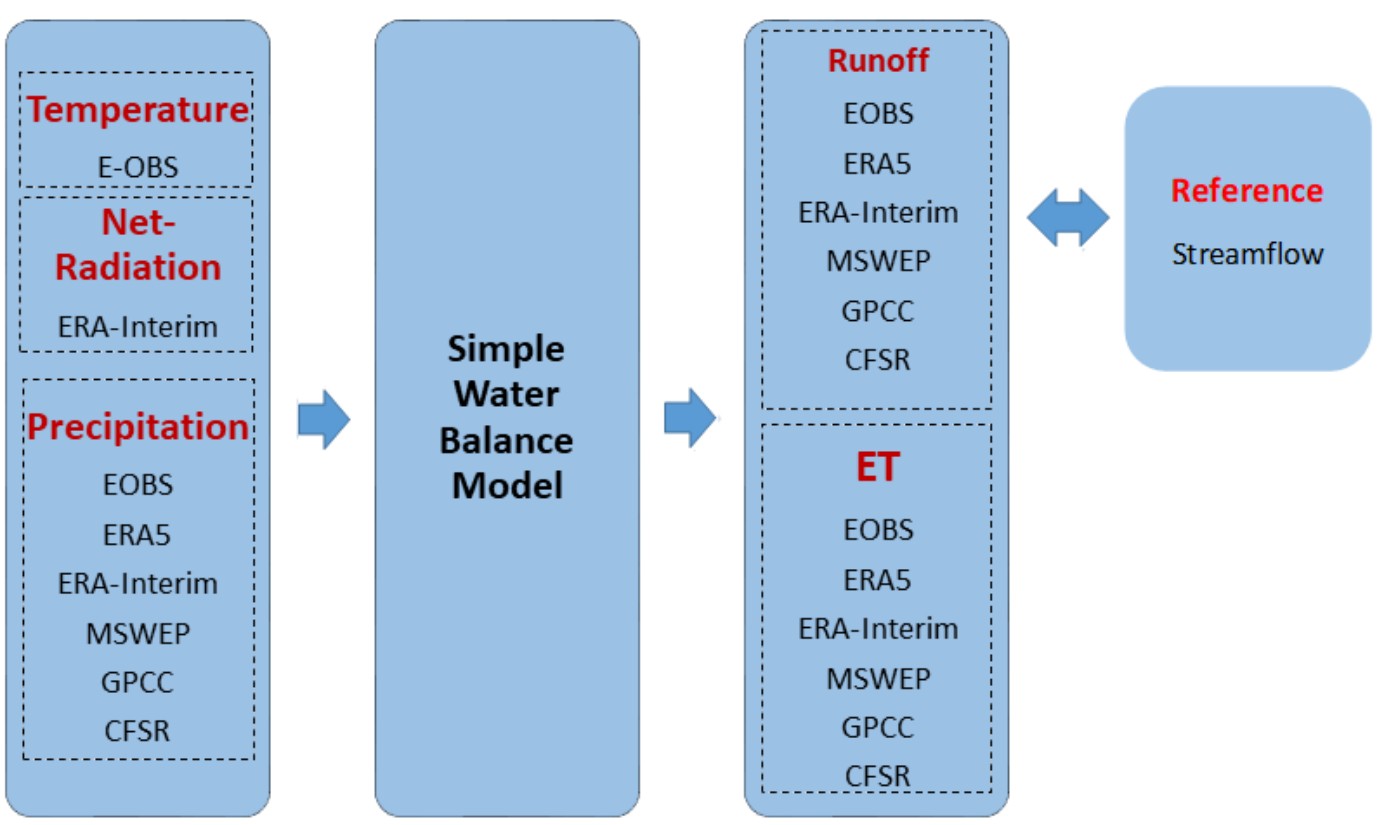

**Figure 2: Overview of the modelling approach. The SWBM model is forced with consistent net radiation and temperature data, but six different precipitation datasets. The obtained runoff and evapotranspiration are assessed in terms of the variability between the simulations. The performance of the runoff simulations is determined against streamflow observations.**

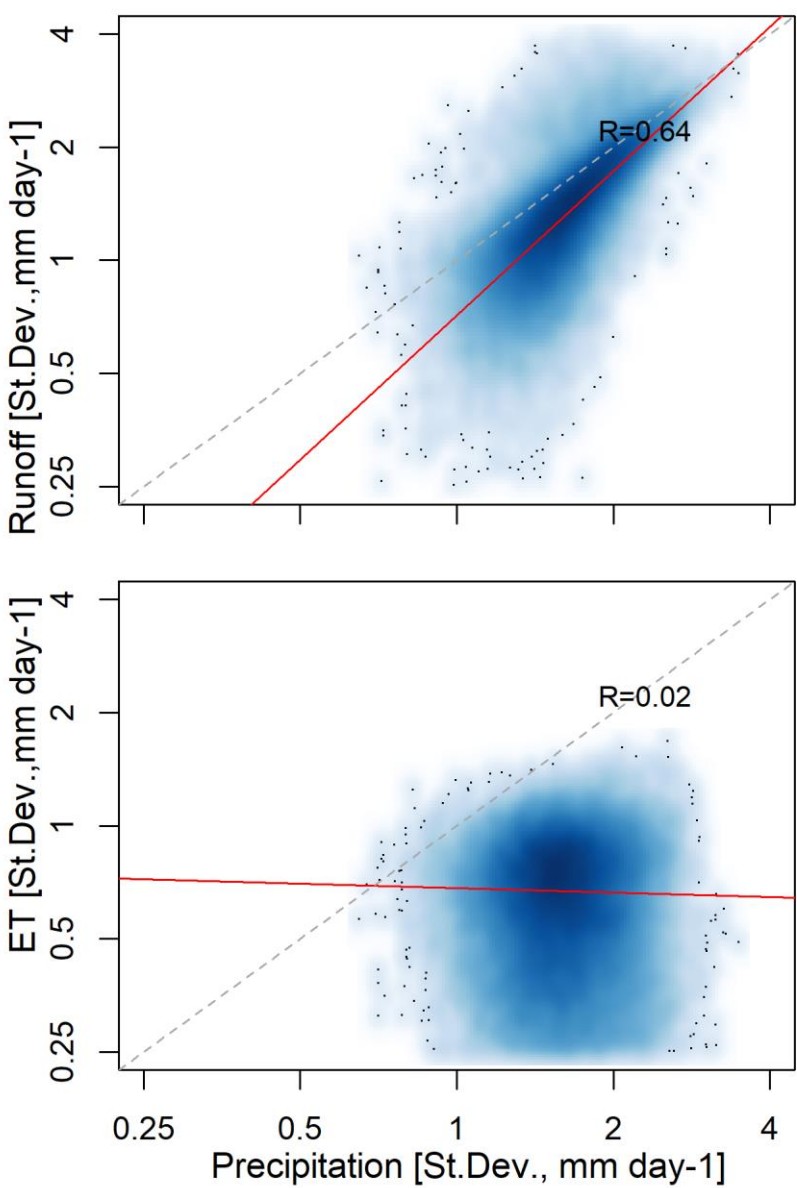

**Figure 3: Propagation of precipitation uncertainty into the runoff and ET simulations. Standard deviations are computed across the precipitation estimates and resulting runoff and evapotranspiration values. This is done at every grid cell and every month between May and September. Red lines indicate linear regression lines. Note that a log-log scale is used.**

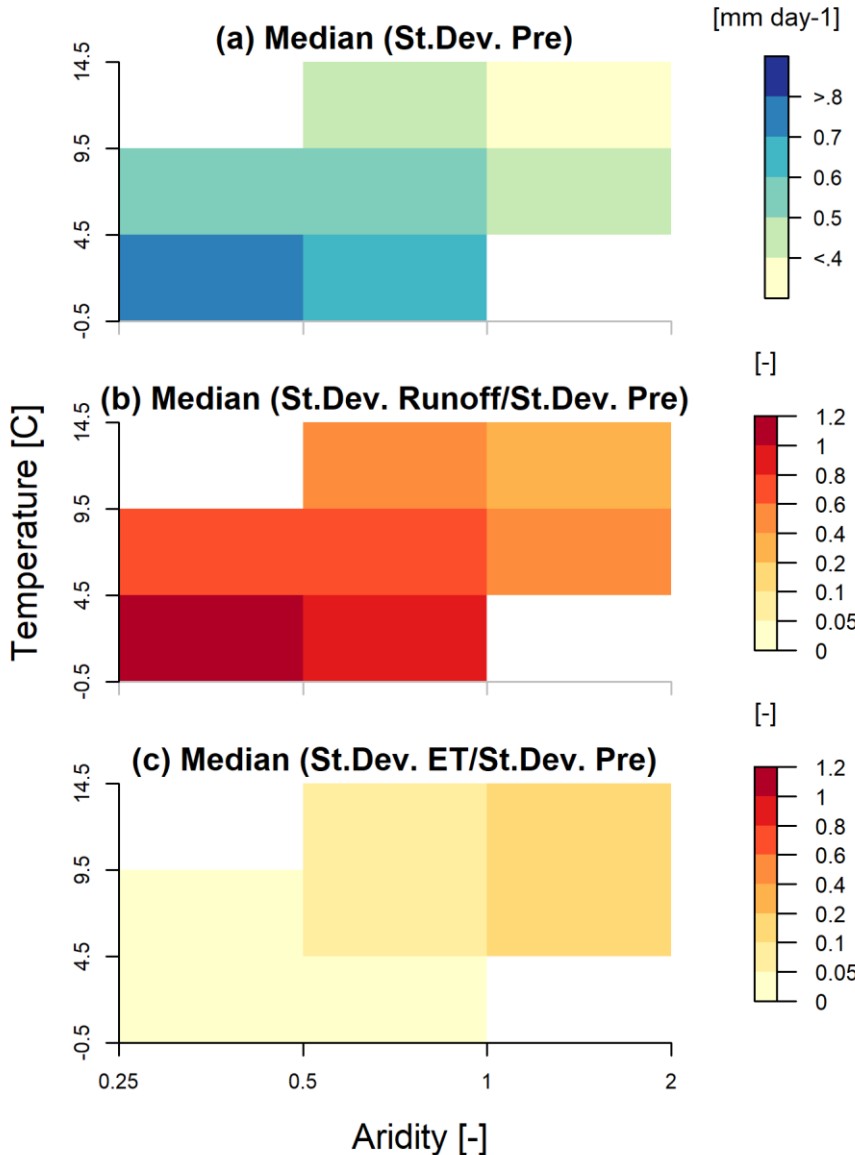

**Figure 4: Climate-dependent propagation of precipitation uncertainty into runoff and ET. a) standard deviation across precipitation products, b) and c) relative standard deviation of resulting runoff and ET simulations with respect to that of precipitation, respectively.**

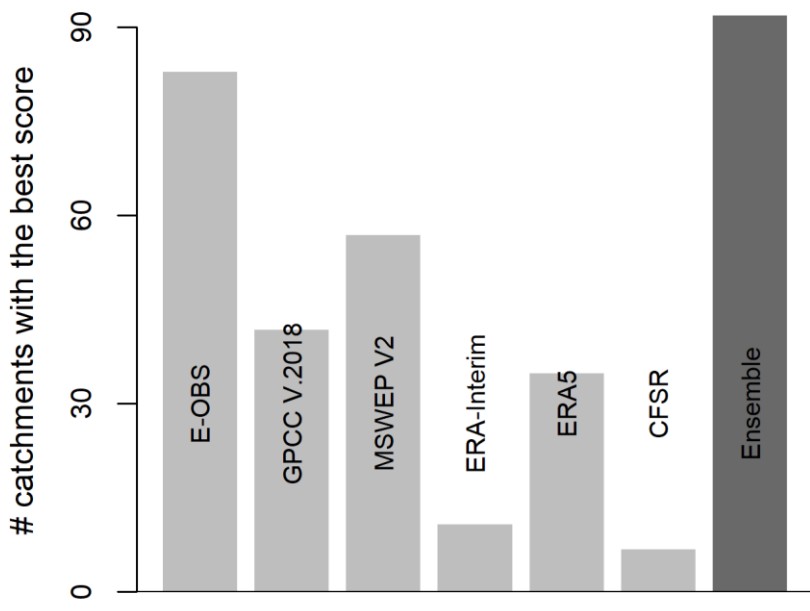

**Figure 5: Number of catchments where each precipitation product yields the best agreement with runoff observations (May-September). Multiple data products can be best-performing at a catchment since they are ranked based on a merged score by combining anomaly correlation and absolute error.**

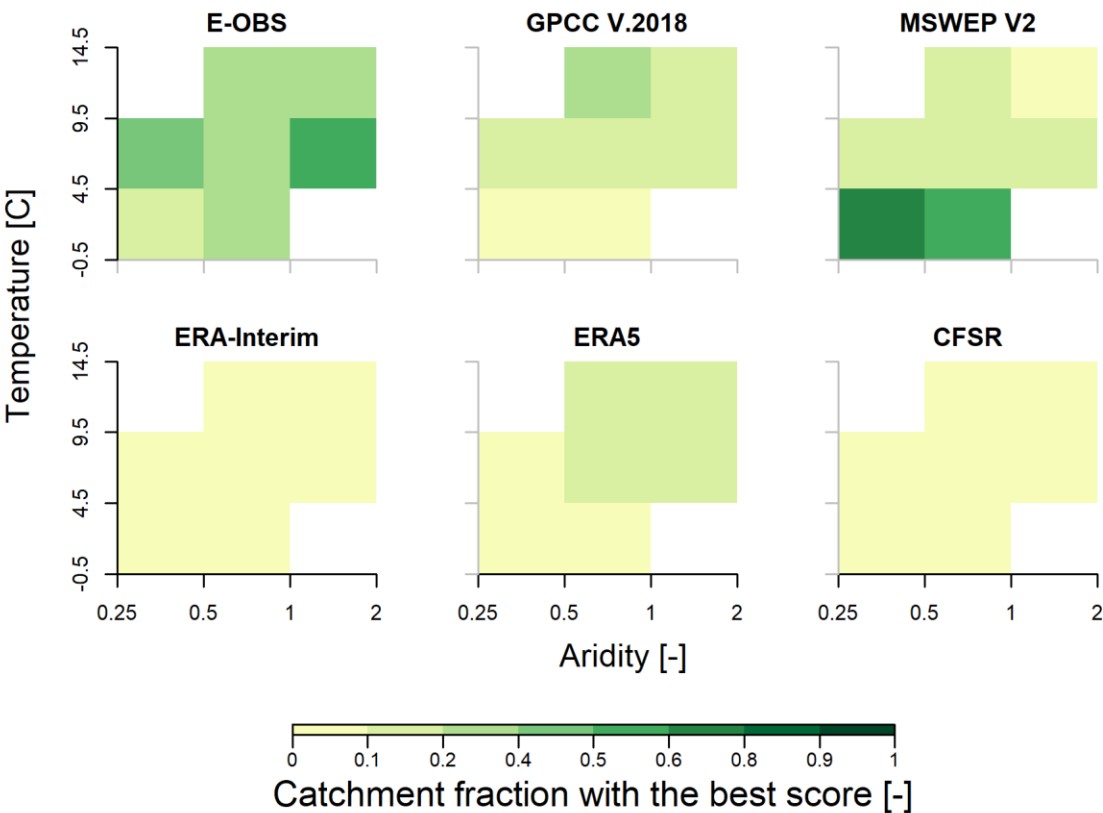

**Figure 6: Runoff-based performance of precipitation products across climate regimes. Colors refer to the percentage of catchments within each box recognized as the best performance based on anomaly correlation and absolute bias during May-September.**