# Peer review of "Climate-dependent propagation of precipitation uncertainty into the water cycle"

_Hydrology and Earth System Sciences, 2019_

## Referee Comment (RC1) · Anonymous Referee #1 · 14 Feb 2020

Summary:

The manuscript has evaluated different precipitation products to understand the uncertainty propagation into the water cycle, specifically streamflow and evapotranspiration. The manuscript evaluated the different products by forcing a lumped model with the different precipitation products and evaluating the outputs, streamflow and evapotranspiration in >200 catchments across Europe.

Overall I think the paper has a good language style and is rather easy to read. I think the study has to be improved in elaboration of the methodology and discussion on different assumptions and how it performs and complements other comparative studies. At this stage I do not believe this study is easily reproducible and this should be the aim of the methodology to a certain extent.

[Figure]

General comments:

In the introduction I would also include existing studies that compare precipitation products directly or indirectly and show explicitly need for this gap that you are filling.

I would include a discussion on how the results from this study performs with the existing studies you will mention in the introduction with comparative investigations. The discussion should also include some assumptions and why these where made and how you justify them with references to literature.

Evaluation or validation of the ET simulations with respect to the available gridded ET datasets may be quite interested to see. Several studies exist where different ET products have been compared for Europe at the basin scale as well as at the European scale and would be valuable to see what might come from this comparison to a 'reference'.

Which calibration method was used, what specific software or was it manual? There should be a certain level of reproducibility possible using the methodology described currently this is not possible as many things are not mentioned. Please elaborate your methodology to include for example:

- Description of how your catchments were selected

- A more detailed description of the model used or where we can find this description

- A description of the model setup, show a schematic of the model architecture.

- A description of the calibration methodology

o What method was used

o which parameters were adjusted

o maybe a map with NSE results from the different catchments. . .

Are your input precipitation datasets open accessible and available? Show in your table

Interactive
comment

1 where people can access these datasets.

It would also be a good idea to list the catchments and some of their details in the supplementary notes so that readers can identify these catchments. Where did you get the data for streamflow (GRDC?) This needs to be mentioned.

Maybe it would be interesting to see results of an ensemble precipitation product that could be used to possibly adjust for the differences in the different products. This could result in overall better performance. It would be nice addition to add an ensemble of your products and see how this performs against the individual products.

Specific comments:

P2L65 – what is the difference between this version of the model you are using and the one you describe initially?

P3L87 . . .'temperature is derived. . ."

P3L99 which month do the points represent? The same for each catchment or different? How did you choose the month?

Calibration is done for model forced with E-OBS precipitation data and you find the model forced with E-OBS data to be the most accurate when comparing streamflow simulated and observed results. Even though you conduct a calibration with small differences in the outputs I think it is important to compare the streamflow results from the second calibrated model (forced with GPCC precipitation data) forced with all precipitation products with observed streamflow results to see if you get similar rankings across catchments.

In the results section, a more detailed interpretation of the resulting graphs need to be made. I think it would be very nice here to show a graph with your NSE for all catchments (map with the values where your catchments are). Maybe then you can also group the results into good, medium and poor performance.

P3L101 you say there is a strong relationship between precipitation and runoff but the R2 is only 0.39. Does this show a strong relationship?
* * *

---

## Referee Comment (RC2) · Anonymous Referee #2 · 11 Mar 2020

[referee-annotated manuscript omitted]

---

## Author Comment (AC1) · 8 Apr 2020

We are thankful to all reviewers for their valuable and constructive feedback which helped us to improve the manuscript. In response, aside from several minor corrections, we have introduced the following main changes to the paper:

(1) We have considerably expanded the methodology section to clarify the model characteristics and its calibration process. In addition, we have added more discussion along with the relevant references to compare the outcome of this study with existing comparative precipitation assessments.

(2) Motivated by the reviewers' ideas, we have developed and expanded our analyses in two main directions by introducing; i) daily-scale results to compare the propagation of rainfall uncertainty across different time scales, and ii) a mean ensemble of the individual precipitation products to explore its potential to reduce uncertainty in runoff simulations.

The results show that our findings on the climate-dependent propagation of precipitation uncertainty are valid across daily and monthly time scales, and that the mean precipitation ensemble yields runoff simulations which agree better with observations than for any individual precipitation product.

Please note that we additionally corrected (i) Figures 3 and 4 to show results for May-September as indicated in the caption, while previously it was erroneously April-October, and (ii) the considered catchments to be consistently determined through the criterion of NSE(runoff)>0.36, resulting in a slight increase in the number of considered catchments.

**Reviewer #1**

Summary:

The manuscript has evaluated different precipitation products to understand the uncertainty propagation into the water cycle, specifically streamflow and evapotranspiration. The manuscript evaluated the different products by forcing a lumped model with the different precipitation products and evaluating the outputs, streamflow and evapotranspiration in >200 catchments across Europe.

Overall I think the paper has a good language style and is rather easy to read. I think the study has to be improved in elaboration of the methodology and discussion on different assumptions and how it performs and complements other comparative studies. At this stage I do not believe this study is easily reproducible and this should be the aim of the methodology to a certain extent.

A1: We thank the reviewer for notifying the fluency of the manuscript and have considered the points as mentioned in the following.

General comments:

In the introduction I would also include existing studies that compare precipitation products directly or indirectly and show explicitly need for this gap that you are filling. I would include a discussion on how the results from this study performs with the existing studies you will mention in the

introduction with comparative investigations. The discussion should also include some assumptions and why these where made and how you justify them with references to literature.

A2: We have added more references and put our results in context of earlier studies throughout the manuscript:

- Results and discussion (lines 123-126):
"It is related to the fact that most of the considered catchments are located in relatively wet climate (aridity<1) such that soils are often saturated, triggering a rather direct runoff response to precipitation (Ghajarnia et al., 2020). Also, in these climate regimes ET is typically energy-controlled rather than water-controlled (Koster et al., 2004, Pan et al., 2019; Zheng et al., 2019; Denissen et al., 2020), leading to the observed low sensitivity of ET to precipitation (uncertainty)."
- Results and discussion (lines 185-188):
"The weaker performance in cold climate, which is also present in the case of E-OBS, might be related to smaller gauge network density, and more complex topography in colder areas (Beck et al., 2017b; Ziese et al., 2018)."
- Conclusions (lines 206-208):
"Thereby the products differ mostly with respect to the temporal dynamics rather than the overall amount of precipitation (Sun et al., 2018; Fallah et al., 2020)."
- Conclusions (lines 224-225):
"Another important outcome of our analyses is that ET simulations are mostly insensitive to precipitation uncertainty in European climate, confirming previous studies (Bhuiyan et al., 2019; Zheng et al., 2019)."

Evaluation or validation of the ET simulations with respect to the available gridded ET datasets may be quite interested to see. Several studies exist where different ET products have been compared for Europe at the basin scale as well as at the European scale and would be valuable to see what might come from this comparison to a 'reference'.

A3: We thank the reviewer for this interesting suggestion. Please note, however, Fig.3 shows that there is no strong relationship between precipitation inputs and ET simulations. Because of this, we cannot re-do the analysis from Figure 5 for ET. Similarly, we also cannot assess the validity of existing ET datasets. To clarify this point, we have updated the manuscript:
- Results and discussion (lines 153-154):
"This is, however, not possible in the case of ET due to the lacking relationship between ET and the precipitation forcing in our study region (Figure 3)."
- Conclusions (lines 225-228):
"However, in warmer and drier regions such as the Middle East, Central North America or Australia, the link between ET and precipitation should be stronger. Wherever available in these regions, ET measurements can and should be used for indirect evaluation of large-scale precipitation products to complement the results in this study where we focused more on comparatively wet regions."

Which calibration method was used, what specific software or was it manual? There should be a certain level of reproducibility possible using the methodology described currently this is not possible as many things are not mentioned. Please elaborate your methodology to include for example:

- Description of how your catchments were selected

- A more detailed description of the model used or where we can find this description

- A description of the model setup, show a schematic of the model architecture.

- A description of the calibration methodology

o What method was used

o which parameters were adjusted

o maybe a map with NSE results from the different catchments.

Are your input precipitation datasets open accessible and available? Show in your table 1 where people can access these datasets.

A4: Addressing the reviewers' comments, more information is added regarding our data and methodology, in particular to more comprehensively describes the selection of catchments, the model, and the calibration method. Also, the access links to the precipitation datasets are provided in the acknowledgment section; all data are publically accessible.

- Data and methodology (lines 69-73):
"We use here the model version introduced by Orth and Seneviratne 2015 which is adapted to the daily time scale by addition of a streamflow recession parameter and an implicit form of the water balance equation. Note that the basic concept and the governing equations of runoff and ET formation used here are well established and employed in many similar conceptual models, such as HBV (Bergström 1995; Orth and Seneviratne, 2015)."
- Data and methodology (lines 101-103):
"Note that we perform only calibration of the model, and no validation. This is because we focus on the influence of the precipitation forcing on the modelled runoff performance, and not on the simulation capacity of the model outside training conditions which has been shown in previous studies (e.g. Orth et al. 2015)."
- Data and methodology (lines 89-94):
"The simple water balance model employed in this study includes six adjustable parameters: water-holding capacity, inverse streamflow recession time scale, runoff ratio exponent, ET ratio exponent, maximum evaporative fraction, and a snow melting parameter (as in Orth and Seneviratne 2015, see also Table S1). For model calibration, 500 parameter sets are tested which are randomly sampled from the entire parameter space using Latin Hypercube Sampling (LHS; McKay et al., 1979). The ranges for each parameter within this parameter space are obtained from O et al., 2020 (see also Table S1). This way, we performed 500 corresponding simulations for each catchment over the entire considered time period 1984-2007."

- Data and methodology (lines 109-112):

"The model parameters are thereby obtained from the above-mentioned calibration using E-OBS precipitation. As this can potentially introduce biases into our results, we additionally calibrated the model using GPCC V.2018 precipitation data to derive alternative parameters with which we re-computed the main analyses."

It would also be a good idea to list the catchments and some of their details in the supplementary notes so that readers can identify these catchments. Where did you get the data for streamflow (GRDC?) This needs to be mentioned.

A5: We have updated the manuscript in lines 84-87 to clarify the origin of the data. For further details on individual catchments, we refer to the original study by Stahl et al. 2010.
- Data and methodology (lines 84-87):
"The streamflow data were collected from the European water archive, national ministries and meteorological agencies and from the WATCH project. These daily data are available for the period 1984-2007… More details on the data and catchments can be found from Stahl et al., 2010."
- Acknowledgment (lines 248-251):
"Further, we are thankful for streamflow data from a dataset compiled by Stahl et al., 2010, who collected data from the European water archive ([http://www.bafg.de/GRDC/], accessed 9 December 2019), from national ministries and meteorological agencies, as well as from the WATCH project ([http://www.eu-watch.org], accessed 9 December 2019)."

Maybe it would be interesting to see results of an ensemble precipitation product that could be used to possibly adjust for the differences in the different products. This could result in overall better performance. It would be nice addition to add an ensemble of your products and see how this performs against the individual products.

A6: Many thanks for this suggestion. We have implemented it and conducted an analysis based on the precipitation ensemble mean. As the reviewer was suspecting, the ensemble yields better agreement between modelled and observed streamflow than any of the individual products. The results are displayed in figures 5 and S7-S10. Discussion on the results can be found in sections:
- Results and discussion (lines 165-167):
"The precipitation ensemble outperforms all individual products, highlighting the usefulness of multi-source and multi-product approaches in the derivation of suitable precipitation datasets for hydrological modelling."
- Conclusions (lines 210-212):
"We further find that the ensemble mean of the considered precipitation datasets outperforms the individual datasets, suggesting that such approaches are promising to obtain more reliable precipitation forcing for hydrological modelling as shortcomings in individual datasets seem to cancel out to some extent when used within an ensemble."

Specific comments:

P2L65 – what is the difference between this version of the model you are using and the one you describe initially?

A7: In the new version, a new streamflow recession parameter and an implicit form of the water balance equation have been implemented.
- Data and methodology (lines 69-71):
"We use here the model version introduced by Orth and Seneviratne 2015 which is adapted to the daily time scale by addition of a streamflow recession parameter and an implicit form of the water balance equation."

P3L87 : : :'temperature is derived: : :"

A8: Corrected (line 105) as "temperature is taken from the E-OBS"

P3L99 which month do the points represent? The same for each catchment or different?

A9: It is pointed at lines 129-130.
 "We compute the median of the standard deviations from catchments within each regime, considering all respective warm season months."

How did you choose the month?

A10: We chose the months "May-September" (now explained at lines 113-114)
 "All analyses are performed during the warm season (May-September) to minimise the impact of snow and ice even though snow melting can locally affect streamflow even in the warm season (Jenicek et al. 2016)."

Calibration is done for model forced with E-OBS precipitation data and you find the model forced with E-OBS data to be the most accurate when comparing streamflow simulated and observed results. Even though you conduct a calibration with small differences in the outputs I think it is important to compare the streamflow results from the second calibrated model (forced with GPCC precipitation data) forced with all precipitation products with observed streamflow results to see if you get similar rankings across catchments.

A11: We confirmed that there is little difference between the E-OBS calibrated and GPCC V.2018 calibrated models in terms of the precipitation data rankings across catchments. Moreover, our main findings regarding the climate-dependent propagation of precipitation uncertainty are not affected by the selection of data for model calibration, as shown in the main figures 4, 5 compared to Supplementary figures S4, S9.

- Data and methodology (lines 109-112):
"The model parameters are thereby obtained from the above-mentioned calibration using E-OBS precipitation. As this can potentially introduce biases into our results, we additionally calibrated the model using GPCC V.2018 precipitation data to derive alternative parameters with which we re-computed the main analyses."
 - Results and discussion (lines 142-148):
"Further, we repeat the uncertainty propagation analysis using (i) model parameters obtained from calibration with GPCC V.2018 precipitation forcing instead of E-OBS precipitation (Figure S4)… We find that Figures S4-S6 show similar patterns as in Figure 4, which confirms that our findings are robust with respect to the methodological approach, particularly in terms of the precipitation dataset employed for model calibration,…"
- Results and discussion (lines 174-177):
"In addition, we re-compute Figure 5 using all months of the year (Fig. S8), and GPCC-derived SWBM parameters (Fig. S9), which both largely confirm the described results. Note that, not surprisingly, model calibration with a particular precipitation product, e.g. E-OBS or GPCCV.2018, leads to the better performance of that respective product."

In the results section, a more detailed interpretation of the resulting graphs need to be made. I think it would be very nice here to show a graph with your NSE for all catchments (map with the

values where your catchments are). Maybe then you can also group the results into good, medium and poor performance.

A12: We thank the reviewer for pointing this out and have added Figure S1 illustrating the NSE values obtained from streamflow observations and simulated runoff time series (line 96). Basically, the results are yielded over catchments with good performance, yet we have done the analyses over catchments with NSE>=0.5 which confirms our findings (lines 142-148).

"Further, we repeat the uncertainty propagation analysis using … (iii) using an NSE limit of 0.5 instead of 0.36 to select catchments where the SWBM is applicable (Figure S6). We find that Figures S4-S6 show similar patterns as in Figure 4, which confirms that our findings are robust with respect to the methodological approach, … the applied NSE threshold to determine the applicability of the model (see also Section 2.3)."

P3L101 you say there is a strong relationship between precipitation and runoff but the R2 is only 0.39. Does this show a strong relationship?

A13: We have toned down the respective paragraph:

- Corrected (lines 120-123):
"Runoff simulations are impacted by precipitation uncertainty while the ET simulations are much less influenced by precipitation uncertainty, as indicated by the regression slope. The clear relationship between runoff and precipitation is in line with previous studies (e.g. Beck et al., 2017a,b; Sun et al., 2018, Blöschl et al., 2019b)."

---

## Author Comment (AC2) · 8 Apr 2020

We are thankful to all reviewers for their valuable and constructive feedback which helped us to improve the manuscript. In response, aside from several minor corrections, we have introduced the following main changes to the paper:

(1) We have considerably expanded the methodology section to clarify the model characteristics and its calibration process. In addition, we have added more discussion along with the relevant references to compare the outcome of this study with existing comparative precipitation assessments.

(2) Motivated by the reviewers' ideas, we have developed and expanded our analyses in two main directions by introducing; i) daily-scale results to compare the propagation of rainfall uncertainty across different time scales, and ii) a mean ensemble of the individual precipitation products to explore its potential to reduce uncertainty in runoff simulations.

The results show that our findings on the climate-dependent propagation of precipitation uncertainty are valid across daily and monthly time scales, and that the mean precipitation ensemble yields runoff simulations which agree better with observations than for any individual precipitation product.

Please note that we additionally corrected (i) Figures 3 and 4 to show results for May-September as indicated in the caption, while previously it was erroneously April-October, and (ii) the considered catchments to be consistently determined through the criterion of NSE(runoff)>0.36, resulting in a slight increase in the number of considered catchments.

**Reviewer #2**

The study setup is nice and well tailored, maybe beside the missing of sub-monthly evaluations of the streamflow. Outcomes are interesting, but not surprize with respect to general expectations.

B1: We thank the reviewer for encouraging comments and detailed suggestions. We have included daily analyses, which also confirm the clear dependency of runoff simulations on the existing uncertainty within the input precipitation dataset and the difference in the uncertainty propagation to runoff and ET simulations across climate regimes. The results are displayed in Figures S3 and S7.

- Results and discussion (lines 140-142):
"In addition to the previous analyses using monthly averaged data, we re-compute Figure 4 using daily data. The results are shown in Figure S3. The similarity between Figures 4 and S3 suggests that our findings on the climate-dependent propagation of precipitation uncertainty are valid across daily and monthly time scales."

- Results and discussion (lines 173-174):
"Repeating the evaluation from Figure 5 with daily data (Figure S7) we find similar results. This suggests that the relative quality of the considered precipitation is comparable across daily and monthly time scales."

I am quite disappointed by the missing information on the calibration and validation of the model. Did I miss a link to previous work with your model?

B2: We thank the reviewer for raising this point. More details on the model calibration/validation are included in (lines 89-94):
"The simple water balance model employed in this study includes six adjustable parameters: water-holding capacity, inverse streamflow recession time scale, runoff ratio exponent, ET ratio exponent, maximum evaporative fraction, and a snow melting parameter (as in Orth and Seneviratne 2015, see also Table S1). For model calibration, 500 parameter sets are tested which are randomly sampled from the entire parameter space using Latin Hypercube Sampling (LHS; McKay et al., 1979). The ranges for each parameter within this parameter space are obtained from O et al., 2020 (see also Table S1). This way, we performed 500 corresponding simulations for each catchment over the entire considered time period 1984-2007."

- Data and methodology (lines 69-73):
"We use here the model version introduced by Orth and Seneviratne 2015 which is adapted to the daily time scale by addition of a streamflow recession parameter and an implicit form of the water balance equation. Note that the basic concept and the governing equations of runoff and ET formation used here are well established and employed in many similar conceptual models, such as HBV (Bergström 1995; Orth and Seneviratne, 2015)."
- Data and methodology (lines 101-103):
"Note that we perform only calibration of the model, and no validation. This is because we focus on the influence of the precipitation forcing on the modelled runoff performance, and not on the simulation capacity of the model outside training conditions which has been shown in previous studies (e.g. Orth et al. 2015)."

See all comments in the attached PDF.

Please also note the supplement to this comment:

https://www.hydrol-earth-syst-sci-discuss.net/hess-2019-660/hess-2019-660-RC2-

supplement.pdf

B3: Many thanks for the detailed comments. We have updated the manuscript following the reviewer's suggestion. Clarifications and additions inserted in response to your comments in the pdf have been highlighted in yellow for better traceability. Further, we particularly thank the reviewer for suggesting the inclusion of the SM2RAIN dataset into our analyses. As an independent and novel dataset, this would have been a valuable addition to our analyses. However, we decided not to use it as it does not cover the investigated time period 1984-2007, and the data gaps constitute a problem for application in hydrological modelling requiring gap-free data, while developing a suitable gap-filling approach was beyond the scope of this analysis.

---

## Author Response (AR2)

**Editor Decision: Publish subject to minor revisions**

Comments to the Author:

Dear Authors

Thank you for your efforts in addressing the reviewer comments. Both are now happy for the paper to proceed to publication, subject to some minor revisions.

Many thanks for acknowledging our efforts in improving the manuscript. We have addressed the remaining minor points in the manuscript and replied accordingly in the responses below.

**From Reviewer 1**

Please consider and respond to these comments:

P3L97 – NSE<0.36. Why this number? Maybe you could justify the use of this threshold? I see you do this later to a certain extent but I would still like to know why you chose this number. Is it random or was there a reason?

Actually the references regarding NSE>0.36 are already specified in the manuscript. However, we have changed their location to make it clear that they are related to the NSE threshold (lines 97-99).

"In addition, any catchments with NSE<0.36 in the case of the best parameter set were disregarded from the further analyses. This NSE threshold for the catchment selection is adopted from Motovilov et al. (1999) and Moriasi (2007)."

P4L141 – I would suggest moving figure S3 into the main article as you are comparing figure 4 and S3 within your results discussion. This would make it easier for the reader to follow. Otherwise, I am happy they have taken the advise that was given and I find the paper much more structured and informative.

Personally, I am not sure that S3 should move to the main body of the manuscript as this is not the only Supplementary figure you compare to other figures, but please do consider the query and amend if you feel otherwise.

We have considered this point and prefer to keep the structure as is.

**From Reviewer 2**
a) Can you include in the supplement the formulation of the new "streamflow recession parameter and an implicit form of the water balance equation." and demonstrate its benefits as compared to the "model version introduced by Orth and Seneviratne 2015".

We think including the equation and demonstrating the benefits is beyond the scope of our study. Instead, addressing the reviewers' comments, we have added some more details and explanations on the streamflow recession parameter and on the validation results for the SWBM in Orth and Seneviratne (2015) (lines 68-71).

"We use here the model version introduced by Orth and Seneviratne (2015) in which the original model is adapted to the daily time scale by addition of an implicit form of the water balance equation and a streamflow

recession parameter which enables streamflow that is delayed with respect to the respective precipitation event. Please refer to Orth and Seneviratne (2015) for the relevant model equations and validation results."

b) Nice that you explain how you calibrate the model. Your sentence on line 101: "Note that we perform only calibration of the model, and no validation." is for me a sign of poor relevance of all outcomes coming afterwards. I would really expect an effort of independent validation.

Please consider and respond to these comments. Regarding the validation component, I do think that you could provide a stronger motivation through the focus on precipitation influence and perhaps, by bringing some of the comments around previous validation (Line 197-199) into the paragraph at Line 101
Regards
Graham Jewitt

We have updated the manuscript in lines 102-105 to include more information on the previous validation of the model.

[revised manuscript text omitted]